# A Novel Monoclonal Antibody Targeting Cancer-Specific Plectin Has Potent Antitumor Activity in Ovarian Cancer

**DOI:** 10.3390/cells10092218

**Published:** 2021-08-27

**Authors:** Samantha M. Perez, Julien Dimastromatteo, Charles N. Landen, Kimberly A. Kelly

**Affiliations:** 1Department of Biomedical Engineering, University of Virginia, Charlottesville, VA 22908, USA; smp7ya@virginia.edu (S.M.P.); ju.dimastro@gmail.com (J.D.); 2ZielBio, Inc., Charlottesville, VA 22903, USA; 3Department of Obstetrics and Gynecology, Division of Gynecologic Oncology, University of Virginia, Charlottesville, VA 22908, USA; cl3nj@hscmail.mcc.virginia.edu

**Keywords:** plectin, monoclonal antibody, ovarian cancer, JAK2/STAT3 signaling, cisplatin, combination therapy

## Abstract

Cancer-specific plectin (CSP) is a pro-tumorigenic protein selectively expressed on the cell surface of major cancers, including ovarian cancer (OC). Despite its assessable localization, abundance, and functional significance, the therapeutic efficacy of targeting CSP remains unexplored. Here, we generated and investigated the anticancer effects of a novel CSP-targeting monoclonal antibody, 1H11, in OC models. Its therapeutic efficacy as a monotherapy and in combination with chemotherapy was evaluated in vitro using two OC cell lines and in vivo by a subcutaneous ovarian cancer model. 1H11 demonstrated rapid internalization and high affinity and specificity for both human and murine CSP. Moreover, 1H11 induced significant and selective cytotoxicity (EC_50_ = 260 nM), G0/G1 arrest, and decreased OC cell migration. Mechanistically, these results are associated with increased ROS levels and reduced activation of the JAK2-STAT3 pathway. In vivo, 1H11 decreased Ki67 expression, induced 65% tumor growth inhibition, and resulted in 30% tumor necrosis. Moreover, 1H11 increased chemosensitivity to cisplatin resulting in 60% greater tumor growth inhibition compared to cisplatin alone. Taken together, CSP-targeting with 1H11 exhibits potent anticancer activity against ovarian cancer and is deserving of future clinical development.

## 1. Introduction

In 2008, Kelly et al. pioneered a phage-display-based functional proteomic approach to identify a subset of proteins with aberrant cell-surface localization in cancer. Most notably amongst these is plectin, which was ubiquitously expressed on 100% of cancer tissues analyzed [1,2]. Since then, multiple studies have further demonstrated that cancer cells have an altered cell-surface proteome, representing a highly attractive class of proteins for targeted therapeutics and immunotherapy [3,4,5,6]. Plectin is a 500kDa cytolinker protein localized to the cytoplasm in pre-malignant tissue but found to be critically mislocalized to the cell surface of pancreatic ductal adenocarcinoma (PDAC) [1,2,7]. Further characterization revealed ovarian cancer, bile duct cholangiocarcinoma, lung adenocarcinoma, lung squamous cell carcinoma, and head and neck cell carcinoma to be positive for cancer-specific plectin (CSP) expression [8,9,10]. Strikingly, Dasa et al. revealed that >90% of serous ovarian cancer (OC) patient samples were positive for CSP [9]. Altogether, CSP+ cancers account for over half of all cancer-related deaths in the United States, making insights into CSP as a therapeutic target that is highly clinically relevant [11]. Given CSP’s accessible localization and abundant expression, it has been used as a cancer biomarker [2,12,13,14,15] as well as a target for directed therapies in ovarian [9] and pancreatic cancer [16,17,18]. However, no previous study has evaluated the therapeutic efficacy of CSP-targeted immunotherapy.

Plectin is a pro-tumorigenic protein whose endogenous suppression by shRNA or siRNA inhibited proliferation, migration, and invasion in vitro and reduced tumor volume and metastases in vivo [7,19,20,21,22,23]. Previously we demonstrated that conferring CSP expression on cells devoid of CSP resulted in increased cell migration and invasion [7]. Moreover, CSP has been identified as a cancer stem cell biomarker. Isolated CSP+ cells had increased clonogenicity and migration compared to CSP cells from the same cell line [10]. CSP’s functional significance has revealed high plectin expression as an indicator for poor prognosis in CSP+ cancers such as lung adenocarcinoma and head and neck squamous cell carcinoma [8,10,19]. Interestingly, in ovarian cancer, a proteomic study investigating makers of peritoneal metastasis found that plectin secretion was enriched during cancer–peritoneal interaction [24]. Additionally, plectin expression was significantly elevated in mice with recurrent disease after initial paclitaxel treatment [25]. These findings demonstrate the functional importance of CSP in cancer, prompting us to hypothesize that CSP could be a clinically relevant therapeutic target.

In this study, we evaluate 1H11′s efficacy in treating CSP + OC. OC accounts for approximately 150,000 annual deaths worldwide, making it the deadliest gynecologic malignancy [26]. This is partly due to ~70% of patients already having advanced-stage disease with wide peritoneal metastasis at the time of diagnosis, for which the 5-year survival rate is only ~30% [11,27]. Under current OC treatment options, 70% of advanced patients will experience disease recurrence and ultimately succumb to the chemoresistant disease [28]. Therefore, new treatment modalities need to be integrated into OC treatment strategies to improve patient survival. The high affinity, specificity, and favorable pharmacokinetics of monoclonal antibodies (mAbs) make them an ideal therapy for improving clinical outcomes and tolerability of OC treatment. Moreover, CSP’s bioavailability and cancer-specific expression pattern optimally position anti-CSP mAb targeting to have low predicted side effects. Aside from anti-VEGF and PARP inhibitors, targeted therapies have shown limited efficacy in OC [28,29]. This study demonstrates the therapeutic utility of targeting noncanonical cell surface proteins, opening the door to this class of proteins as novel therapeutic targets.

Here, we generated and characterized a novel CSP-targeting antibody, 1H11, evaluated its therapeutic efficacy as a monotherapy and in combination with chemotherapy, as well as investigated 1H11′s anticancer mechanism in ovarian cancer.

## 2. Materials and Methods

### 2.1. Patient Survival Analysis

The prognostic value of plectin (Affymetrix probe ID = 201373_at) in ovarian cancer was analyzed using online survival software kmplot (http://kmplot.com/analysis/; accessed on 31 May 2021), which pools gene expression and patient survival data of 1287 ovarian cancer patients from Gene Expression Omnibus (GEO) and The Cancer Genome Atlas (TCGA) [30,31]. The stratification into high- and low- expressing patients was determined by “auto select best cutoff.” The software calculated the hazard ratio (HR) with 95% confidence intervals and log-rank *p* values.

### 2.2. Cell Culture and Reagents

Murine PDAC cell line, Han14.3, was obtained from Nabeel Bardeesy (Harvard Medical School, Massachusetts General Hospital, Boston, MA, USA). Human OC cell lines, OVCAR8 and SKOV3, and healthy fallopian tube cells, FT132, were gifted by Jill Slack Davis (University of Virginia, Charlottesville, VA, USA). Han14.3, SKOV3, and OVCAR8 cells were grown in RPMI supplemented with 10% (*v*/*v*) FBS, 1% (*v*/*v*) penicillin-streptomycin, and 1% (*v*/*v*) l-glutamine. FT132 cells were maintained in DMEM supplemented with 10% (*v*/*v*) FBS, 1% (*v*/*v*) penicillin-streptomycin, and 1% (*v*/*v*) l-glutamine. All cells were maintained at 37 °C in 5% CO_2_. Han14.3, OVCAR8, and SKOV3 cells were previously validated to be CSP+, while FT132 cells were shown to be CSP- [7,9]. Cells were routinely tested for mycoplasma using the MycoProbe Mycoplasma Detection Kit (R&D Systems, Minneapolis, MN, USA).

For all experiments, mouse IgG2b (BioCell, Lebanon, NH, USA) was used as an isotype control. Cisplatin (Sigma-Aldrich, St. Louis, MO, USA), olaparib (AZD2461; Sigma-Aldrich, St. Louis, MO, USA), and doxorubicin (Sigma-Aldrich, St. Louis, MO, USA) were used for combinational experiments.

### 2.3. Generation and Characterization of 1H11

Monoclonal antibodies were generated and purified by GenScript (Piscataway, NJ, USA). Five Balb/c mice were immunized with a His-tagged recombinant 32kDa fragment of human CSP (rhCSP*) corresponding to the C-terminal region of plectin, according to GenScript’s proprietary MonoExpress immunization protocol. Isolated splenocytes were then fused to myeloma cells by electrofusion. The supernatant from generated hybridomas was screened for reactivity to rhCSP* via ELISA by Genscript. In-house, the specificity and affinity of 20 lead hybridomas to native CSP was evaluated by ELISA using mAb serum samples against CSP+ and CSP- cells. The lead candidate clone, 1H11, was subcloned by limiting dilution, expanded, and purified with protein A columns by GenScript.

To evaluate binding kinetics, 1H11 was immobilized onto AMC sensors and binding constants were calculated for associate and dissociation to rhCSP* or His-tagged recombinant GST (rhGST) using Octet RED96 system (ForteBio LLC, Fremont, CA, USA). Affinity and specificity were also evaluated by in proteo ELISA in which plates were coated with either rhCSP* or rhGST, blocked with 2% (*w*/*v*) bovine serum albumin (BSA), then incubated with serial dilutions of 1H11. Plates were washed before and after incubation with anti-mouse IgG (HRP) (Abcam, Cambridge, MA, USA), then incubated with TMB (Thermo Fisher Scientific, Waltham, MA, USA) and readout at 650 nm (Molecular Devices Microplate Reader, San Jose, CA, USA). Data were corrected for non-specific binding to rhGST. For cell-binding ELISA assays, cells were blocked with 2% (*w*/*v*) BSA, then incubated with serial dilutions of 1H11 or IgG at 4 °C followed by washes with ice-cold PBS before and after fixing with 4% paraformaldehyde (PFA)/PBS. Cells were then incubated with anti-mouse IgG (HRP) and rinsed before incubation with TMB. The absorbance of each well was measured at 650 nm (Molecular Devices Microplate Reader, San Jose, CA, USA). Data were normalized to cell amount using Janus green staining (Abcam, Cambridge, MA, USA) and corrected for non-specific binding to IgG.

### 2.4. Immunofluorescence Staining for Endosome Markers

OVCAR8 cells were incubated for 30 min at 37 °C in medium containing 1 µM of 1H11 or IgG. Cells were washed with ice-cold HBSS, incubated with Wheat Germ Agglutinin (WGA) conjugated with Alexa Fluor (Thermo Fisher Scientific, Waltham, MA, USA), then washed before and after fixing with 4% PFA/PBS. Cells were blocked with 5% (*w*/*v*) BSA and 0.05% (*w*/*v*) saponin, then incubated overnight with primary antibodies diluted in blocking solution, anti-LAMP1 antibody (Abcam, Cambridge, MA, USA), or anti-EEA1 (Abcam, Cambridge, MA, USA). Cells were then washed before and after incubation with secondary antibodies (anti-mouse IgG (Alexa Fluor 594) (Abcam, Cambridge, MA, USA) or anti-rabbit IgG (Alexa Fluor 647) (Abcam, Cambridge, MA, USA). Finally, the slides were mounted using Prolong^®^ gold antifade mountant with DAPI (Thermo Fisher Scientific, Waltham, MA, USA). Images were acquired using the Zeiss LSM 880 Confocal Microscope (ZIESS, Waltham, MA, USA) at the Advanced Microscopy Facility at the University of Virginia.

### 2.5. Cell Viability

Inhibitory effects were evaluated by Sulforhodamine-B (SRB) assay. Cells were incubated with serial dilutions of treatment conditions for 72 h. Then, cell density was quantified as described by Vichai et al. [32]; however, using 0.04% (*w*/*v*) SRB solution in 1% (*v*/*v*) acetic acid instead. The optical density was measured at 540 nm (Molecular Devices Microplate Reader, San Jose, CA, USA). Data are plotted as:(1)$$\%\mathrm{control}\mathrm{growth}=\frac{{\mathrm{OD}}_{\mathrm{sample}}-{\mathrm{OD}}_{\mathrm{Day}0}}{{\mathrm{OD}}_{\mathrm{control}\mathrm{sample}}-{\mathrm{OD}}_{\mathrm{Day}0}}\times 100$$
EC_50_ was calculated by logistic nonlinear regression using Prism 9.0 (GraphPad Software, San Diego, CA, USA). The interaction of 1H11 and anticancer agents was evaluated by calculation of the combination index (CI) using the CompuSyn software as described by [33]. CI values at 50%, 90%, and 95% growth inhibition are reported. The ranges of CI are defined as <0.1 = very strong synergism, 0.1–0.3 = strong synergism, 0.3–0.7 = synergism, 0.7–0.85 = moderate synergism, 0.85–0.90 = slight synergism, 0.90–1.1 = additive effect, and >1.0 = antagonism.

### 2.6. Cell Migration Assay

Cell migration was determined by wound healing (scratch assay) in which cells were serum-starved overnight, then a scratch with a P200 pipette tip was inflicted on a confluent monolayer. Then, cells were rinsed with HBSS, and fresh serum-free medium containing either 2 µM 1H11 or IgG was added. The wound was imaged for up to 48 h using the EVOS FL imager (Thermo Fisher Scientific, Waltham, MA, USA). Wound size was determined using the MRI Wound Healing macro for ImageJ. Migration rate was calculated as:(2)$$\mathrm{Migration}\mathrm{Rate}=100-\frac{{\mathrm{Wound}\mathrm{Area}}_{T_{48\mathrm{hr}}}}{{\mathrm{Wound}\mathrm{Area}}_{T_{0}}}\times 100$$

### 2.7. Cell Cycle Phase Distribution

Cells were treated for 72 h with 1H11 or IgG, then washed with HBSS and fixed with 70% ethanol overnight. Cells were stained with a solution of 15 µg/mL of propidium iodide (Thermo Fisher Scientific, Waltham, MA, USA) with 0.2 mg/mL of DNAse free RNAse A (Sigma-Aldrich, St. Louis, MO, USA) diluted in 0.1% (*v*/*v*) TritonX-100 in PBS. Cells were detected and analyzed using the BD FACSCalibur (Cytek Biosciences, Fremont, CA, USA) by the Flow Cytometry Core Facility at the University of Virginia. Cell cycle distribution was analyzed by FlowJo Software v10 (Becton, Dickinson, Ashland, OR, USA).

### 2.8. Reactive Oxygen Species (ROS) Detection

Cells were treated with 2 µM 1H11 or IgG for 72 h then the DCFDA/H2DCFDA Assay Kit (Abcam, Cambridge, MA, USA) was used per the manufacturer’s protocol.

### 2.9. Western Blot Analysis

Cells were serum-starved overnight prior to treatment with 125 nM of 1H11 or IgG. Cells were washed with ice-cold PBS, then whole-cell lysates were extracted by RIPA lysis buffer supplemented with protease and phosphatase inhibitors (Thermo Fisher Scientific, Waltham, MA, USA), and protein concentration was measured by bicinchoninic acid assay (Thermo Fisher Scientific, Waltham, MA, USA). Equal amounts of protein were separated using precast 4–15% tris glycine eXtended polyacrylamide gels (Bio-Rad, Hercules, CA, USA) and transferred onto PVDF membranes (Millipore Sigma, Burlington, MA, USA). After blocking with Odyssey Blocking Buffer (LI-COR, Lincoln, NE, USA), membranes were incubated with relevant primary antibody at 4 °C overnight, followed by washes and incubation with infrared dye-conjugated secondary antibodies (LI-COR, Lincoln, NE, USA). Membranes were imaged on a LI-COR Odyssey Infrared Imager, and densitometry measurements were calculated using Image Studio software v5.4.5 (LI-COR, Lincoln, NE, USA). Primary antibodies used include pJAK2^Tyr1007/1008^ (#3771), pSTAT3^Ser727^ (#9134), p21 (#2947), p27 (#3686) and GAPDH (#2118) from Cell Signaling Technology (Danvers, MA, USA) as well as JAK2 (sc-390539), STAT3 (sc-8019), E-cadherin (sc-8426) and GAPDH (sc-32233) from Santa Cruz Biotechnology (Dallas, TX, USA). Vimentin (ab92547) from Abcam (Cambridge, MA, USA) was also used.

### 2.10. Therapeutic Efficacy of Anti-CSP mAb Treatment In Vivo

Animal experiments were performed with the approval of the Institutional Animal Care and Use Committee (IACUC) at the University of Virginia (Protocol #3731). Athymic nude mice were injected subcutaneously with OVCAR8 cells. Once tumor volume reached ~100 mm^3^, mice were randomized into treatment and control groups (*n* = 6 mice/group). Mice were administered 5 mg/kg of 1H11 or 5 mg/kg IgG. Two mice per group were randomly chosen and harvested after one week of treatment for analysis (data not shown), while the rest were treated for three weeks. For combination studies, three additional groups (*n* = 5 mice/group) were used: (1) 2 mg/kg cisplatin alone, (2) 1 mg/kg 1H11 and 2 mg/kg cisplatin, or (3) 1 mg/kg IgG and 2 mg/kg cisplatin. One mouse per group was randomly chosen and harvested after two weeks of treatment for analysis (data not shown), while the rest were treated for 28 days. All mice were treated intravenously twice weekly throughout the course of the study. Tumor volume (TV) was measured twice a week using calipers and the formula:(3)$${\mathrm{TV}=L\times W}^{2}$$
where L = length and W = width of tumor. Percent tumor growth inhibition (%TGI) was calculated as a readout of treatment efficacy as described in [34]:(4)$$\%\mathrm{TGI}=1-\frac{\left({\mathrm{TV}}_{T,t}/{\mathrm{TV}}_{T,0}\right)}{\left({\mathrm{TV}}_{C,t}/{\mathrm{TV}}_{C,0}\right)}$$
where V_T,t,_ and V_T,0_ are the mean TV of the treatment group at day t and 0, respectively. V_C,t,_ and V_C,0_ are the mean TV of the control group at day t and 0, respectively. Mouse body weight was also recorded during the course of treatment. At the end of the study, mice were euthanized, and major organs and tumor tissue were harvested then fixed with 4% PFA/PBS for paraffin embedding. Serum samples from tumor-bearing mice were collected to measure readouts for kidney and liver function such as blood urea nitrogen (BUN), creatinine, and alanine transaminase (ALT) using the VetTest Chemistry Analyzer (IDEXX Laboratories, Westbrook, ME, USA).

### 2.11. H&E and IHC Analysis

H&E staining was performed by the Histological facility at the Cardiovascular Research Center at the University of Virginia. Sections of 5 µm tissue were deparaffinized and stained using Hematoxylin 1 (Richard Allan Scientific, Kalamazoo, MI, USA) and Eosin (Sigma-Aldrich, St. Louis, MO, USA).

Immunohistochemical staining for plectin, Ki67, E-cadherin, and cleaved-caspase 3 was performed on a robotic platform (Discovery Ultra Staining Module, Roche Diagnostics, Indianapolis, IN, USA) by the Biorepository and Tissue Research Facility at the University of Virginia. Tissue sections (4 µm) were deparaffinized, and a heat-induced antigen retrieval protocol was carried out using Cell Conditioner 1 (Roche Diagnostics, Indianapolis, IN, USA). Endogenous peroxidases were blocked with peroxidase inhibitor (CM1) before incubating the sections with plectin (Abcam, Cambridge, MA, USA; catalog # ab32528), Ki67 (Abcam; catalog # ab16667), E-cadherin (Abcam; catalog # ab40772), or cleaved-caspase 3 (Cell Signaling Technology, Danvers, MA, USA; catalog # 9661). The antigen–antibody complex was then detected using OmniMap anti-rabbit multimer with DISCOVERY ChromoMap DAB Kit (Roche Diagnostics, Indianapolis, IN, USA). For p21 staining, antigen retrieval was performed using the Dako PT Link instrument (Agilent Dako, Santa Clara, CA, USA), then immunohistochemistry was performed on a robotic platform (Agilent Dako, Santa Clara, CA, USA). Endogenous peroxidases were blocked using Peroxidase and Alkaline Phosphatase Blocking Reagent (Agilent Dako, Santa Clara, CA, USA), and p21 (Santa Cruz Biotechnology, Dallas, TX, USA; catalog # sc-6246) was visualized by incubation with EnvisionTM Rabbit Link (Agilent Dako, Santa Clara, CA, USA) followed by incubation with 3,3′-diaminobenzidine tetrahydrochloride (DAB+). All the slides were counterstained with hematoxylin, dehydrated, cleared, mounted, and scanned using the Aperio ScanScope (Leica Biosystems, Buffalo Grove, IL, USA). For each tumor, three tissue sections were analyzed. Image analysis of the percentage of DAB-positive cells was calculated as the proportion of positively stained cells out of the total number of cells per tissue, irrespective of localization using QuPath v.0.2.3 [35]. Similarly, tumor necrosis was detected regardless of localization using QuPath’s pixel classification function [35].

### 2.12. Statistical Analysis

Prism 9.0 (GraphPad Software, San Diego, CA, USA) was used to perform statistical tests. All experiments were formed at least three times with data represented as the mean ± SEM. A two-tailed Student’s *t*-test for two-group comparisons and a one-way ANOVA followed by Tukey’s multiple comparison test were performed for multiple group comparison. Tumor growth curves were analyzed by a mixed-effects model followed by a multiple comparison test. A *p* < 0.05 was considered statistically significant.

## 3. Results

### 3.1. Plectin as a Prognostic Marker in OC

CSP was previously identified to be expressed on the cell surface of high-grade serous carcinoma, the most common OC subtype characterized by ubiquitous TP53 mutations [8,9,28]. A patient survival analysis using the Kaplan–Meier (KM) plotter database, which integrates GEO and TCGA data, revealed that for this subset of patients, high plectin (PLEC) mRNA expression was significantly associated with worse overall survival (OS) (*n* = 493, HR = 1.4, 95% CI: 1.09–1.79, *p* = 7.8 × 10^−3^; Figure 1A). Plectin mRNA expression was also significantly associated with worse post-progression survival (PPS) (*n* = 312, HR = 1.69, 95% CI: 1.28–2.24, *p* = 1.8 × 10^−4^; Figure 1B). Moreover, an analysis of the differential expression levels of plectin mRNA between serous carcinoma and normal tissue via Oncomine revealed that plectin mRNA is significantly overexpressed in serous ovarian cancer tissues (Figure 1C). These results suggest plectin plays an important role in OC tumorigenesis. Thus, we decided to evaluate if therapeutic targeting of CSP could serve as a novel anticancer treatment modality in OC.

### 3.2. Generation and Characterization of 1H11

Due to the high specificity, affinity, and favorable pharmacokinetics of monoclonal antibodies, an immunologic approach was used for the therapeutic targeting of CSP. Anti-CSP mAbs were developed using a purified recombinant fragment of human CSP (rhCSP*) to immunize mice. rhCSP* corresponds to a subsection of CSP displayed on the cell surface of cancer cells, whose binding causes internalization [1,9]. The generated mAb clones were screened for affinity and specificity to rhCSP* and native CSP on cells. 1H11 was identified as the lead candidate clone and chosen for further validation (Figure 2A).

To determine the binding affinity and specificity of 1H11 to CSP, binding kinetics experiments and in proteo ELISAs were performed with rhCSP*. They revealed 1H11 to have similar binding affinities with dissociation constants (K_D_) of 7.04 nM and 0.132 nM, respectively (Figure 2B,C). To evaluate whether 1H11 recognizes both murine and human CSP, cell-binding ELISA assays were performed. 1H11 showed high binding affinity to two CSP+ OC cell lines, SKOV3 and OVCAR8 cells, with K_D_s of 22.3 nM and 21.1 nM, respectively (Figure 2D). Moreover, 1H11 cross-reacted with murine CSP as seen by specific binding to CSP+ murine Han14.3. cell line with a nearly identical K_D_ of 21.9 nM (Figure 2E) comparable to CSP on human cells [7]. A protein analogy search showed 97.5% homology between human and murine rhCSP*. Thus, 1H11 demonstrates high binding affinity and specificity for rhCSP* and both murine and human native CSP.

Due to 1H11 targeting the fragment of CSP known to cause cell internalization, we investigated if 1H11 is endocytosed. OVCAR8 cells were incubated with 1H11 or IgG for 30 min, then co-stained for early endosome antigen 1 (EEA1) or lysosomal-associated membrane protein 1 (LAMP-1), a late endosomal biomarker. 1H11 showed a high association with both early and late endosome biomarkers compared to IgG suggesting 1H11 is rapidly internalized after binding (Figure 2F,G). 1H11 was contained within the vesicles.

### 3.3. Targeting CSP Inhibits OC Cell Viability and Migration and Induces G0/G1 Arrest

To determine the in vitro efficacy of 1H11, its effects on OC cell viability, cell-cycle phase distribution, and migration were evaluated. The cytotoxic effects of 1H11 were determined by SRB assay. Results indicate that treatment with 1H11 decreased OC cancer cell viability in a dose-dependent manner in both OVCAR8 and SKOV3 with an EC_50_ of 260 nM (Figure 3A,B). Importantly, 1H11 did not display inhibitory effects in normal fallopian tube cells, FT132, which have negligible CSP expression (Figure 3C) [9]. Next, the cell cycle distribution of OVCAR8 after treatment with 1H11 (0–125 nM) for 72 h was investigated by flow cytometry. Results demonstrated that 1H11 caused a significant increase in the percentage of cells in G0/G1 in a dose-dependent manner. CSP-targeting caused the proportion of cells in G0/G1 to increase from 48% to 59%, while the distribution of cells incubated with IgG did not change. In accordance, a decrease in the percentage of cells in the S-phase was also seen (Figure 3D). G0/G1 cell cycle arrest was also seen in SKOV3 but not in FT132 cells (Appendix A). Thus, these results indicate that 1H11 induces cytotoxicity and G0/G1 arrest in a CSP-specific manner.

Next, the impact of CSP targeting on OC cell migration was investigated by scratch assays. After 48 h of treatment, 1H11-treated SKOV3 cells had 32% of their wound open compared to 6% in IgG treated cells (Figure 4A,B). Similarly, the migration rate was also significantly inhibited by 1H11 (67%) compared to IgG (93%) (Figure 4C). 1H11 also reduced migration in OVCAR8 cells leading to 76% of the wound remaining open as opposed to 63% in IgG treated cells (Figure 4D,E). As expected, the migration rate was also significantly hindered by 1H11 (24%) compared to control (37%) (Figure 4F). These results are consistent with reports showing SKOV3 to be more migratory than OVCAR8 [36]. Thus, OC cell migration was significantly suppressed in the presence of 1H11 compared to IgG. Overall, these results demonstrate that 1H11 has potent anticancer effects on OC cells in vitro.

### 3.4. H11 Induces ROS Accumulation and Suppression of JAK2-STAT3 Signaling

Previous studies have demonstrated that plectin targeting results in ROS accumulation [37,38,39]. Thus, we sought to investigate if 1H11 caused a similar effect. As shown in Figure 5A, treatment with 1H11 resulted in an increase in ROS compared to IgG. ROS accumulation has been reported to dampen proliferative pathways. In particular, increased ROS levels have been shown to suppress the JAK2-STAT3 pathway in OC [40,41,42]. Thus, to further investigate 1H11′s molecular mechanism, we evaluated if treatment modulates the JAK2-STAT3 pathway by Western blot. OVCAR8 and SKOV3 cells treated with 1H11 showed decreased levels of JAK2 phosphorylation compared to IgG-treated cells (Figure 5B and Appendix A). Moreover, 1H11 also inhibits the activation of STAT3, a critical downstream effector of JAK2 (Figure 5C and Appendix A). Once phosphorylated, STAT3 regulates the expression of key proteins involved in cell-cycle progression and migration. In particular, suppression of STAT3 has been reported to increase expression of cyclin-dependent kinase inhibitors, p21 and p27, resulting in induction of G0/G1 arrest [43]. Results indicate that 1H11 treatment for 72 h resulted in a significant increase in p21 and p27 protein levels (Figure 5D). In OC, STAT3 has also been implicated in regulating key EMT proteins that regulate motility, E-cadherin and vimentin [44,45,46]. 1H11 treatment was associated with the upregulation of E-cadherin and downregulation of vimentin (Figure 5E). These results are consistent with 1H11′s previously observed induction of G0/G1 arrest and decreased migration (Figure 3 and Figure 4). Overall, we demonstrate that 1H11 elevates cellular ROS levels and inhibits the activation of JAK2-STAT3 signaling, resulting in increased expression of anti-proliferative and anti-migratory effectors (Figure 5F).

### 3.5. H11 Treatment Inhibits OC Tumor Growth In Vivo

The therapeutic efficacy of 1H11 on tumor growth was evaluated using a xenograft tumor model by subcutaneously injecting OVCAR8 cells into athymic nude mice. Once tumors reached a volume of 100 mm^3^, mice were treated with 1H11 (5 mg/kg) or IgG (5 mg/kg) twice weekly. Toxicity studies using serum samples after a week of treatment showed no significant changes in readouts for liver (ALT) and kidney (BUN and creatinine) function compared to IgG treated mice (Figure 6A). The histology of major organs was evaluated by H&E. No major differences were observed, indicating that administration of 1H11 does not induce major toxic effects (Figure 6B).

After one week of treatment, 1H11-treated mice demonstrated a significant reduction in tumor growth (%TGI = 65%) and tumor regression compared to IgG-treated mice. Throughout the course of treatment, 1H11 continued to significantly delay tumor growth (%TGI > 50%) (Figure 7A,B). No change in mouse weight, skin appearance, or animal behavior was observed over the course of treatment, indicating no overt toxicity (Appendix A). CSP expression was evaluated in tumor tissue samples by IHC staining for plectin. Both 1H11 and IgG-treated tumors demonstrate localization of plectin at the cell membrane (Figure 7C).

IHC staining was used to evaluate protein expression patterns in vivo. Cell proliferation was assessed by staining for Ki67 and p21. Consistent with our results in vitro, 1H11-treated tumors showed a decrease in proliferation as demonstrated by the significant decrease of Ki67 expression and increase of p21 expression. Similar to in vitro, 1H11 treatment increased E-cadherin expression. (Figure 7D). Furthermore, H&E staining revealed a significant increase in tumor necrosis in mice treated with 1H11 (Figure 7E). These results demonstrate that 1H11 induces potent anticancer effects, in part, by decreasing cell proliferation and increasing cell death in vivo.

### 3.6. H11 Increases Chemosensitivity in OC In Vitro and In Vivo

Cytoreductive surgery and combination chemotherapy with platinum and taxane compounds are currently considered the standard OC adjuvant treatment [28]. The prognostic value of plectin mRNA expression status on OS and PPS for patients with the most common OC subtype, serous mutant TP53 tumors, who received platinum-based therapy was evaluated by KM plotter. High expression of plectin mRNA expression was significantly associated with poor rates of OS (*n* = 456, HR = 1.53, 95% CI: 1.18–1.98, *p* = 1.3 × 10^−3^; Figure 8A) and PPS (*n* = 303, HR = 1.74, 95% CI: 1.32–2.3, *p* = 7 × 10^−5^; Appendix A). Thus, we evaluated the therapeutic efficacy of 1H11 in combination with first-line therapy, cisplatin, by SRB assay. In OVCAR8 cells, 1H11 showed strong synergism (CI < 0.3) with cisplatin at a 1:2 molar ratio resulting in a greater than 20-fold increase in potency (Figure 8B,E). Moreover, combinational treatment with 1H11 and PARP inhibitor, olaparib, at a 1:1 molar ratio resulted in a > 10-fold increase in potency with synergism (CI < 0. 1) seen at high effect levels (ED_90_ and ED_95_) (Figure 8C,E). Similarly, 1H11 and doxorubicin dual treatment at a 1:1 molar ratio resulted in a seven-fold increase in potency and a synergistic interaction (CI ≤ 0.14) (Figure 8D,E). These results demonstrate that combinational treatment with 1H11 enhances sensitivity to first-line OC therapies (Figure 8B–E) and 1H11 monotherapy (Figure 3A).

Next, we sought to investigate if the combinational treatment of 1H11 and cisplatin showed augmented anticancer potency compared to cisplatin alone. The tumor volume of athymic nude mice with subcutaneously implanted OVCAR8 tumors was monitored across three treatment groups: (1) 1H11 (1 mg/kg) and cisplatin (2 mg/kg), (2) IgG (1 mg/kg) and cisplatin (2 mg/kg), and(3) cisplatin (2 mg/kg) alone. Mice treated with a combination of 1H11 and cisplatin demonstrated significant suppression in tumor volume compared to cisplatin alone and combinational treatment with IgG (Figure 9A). Strikingly, after one week of treatment, administration of a combination of 1H1 and cisplatin not only induced a significant 50% reduction in tumor growth but also resulted in tumor regression (Figure 9A,B). This trend is an enhancement of our previous observation of the in vivo effects of 1H11 monotherapy (Figure 7A,B). Moreover, tumor weight was significantly lower in mice administered combined treatment with 1H11 (Figure 9C). No changes in body weight were observed across treatment groups (Appendix A). Taken together, our data indicate that 1H11 enhances cisplatin efficacy in vivo using in a xenograft subcutaneous mouse model.

## 4. Discussion

CSP has been revealed as an important regulator of cancer cell proliferation, migration, and invasion, yet few studies have interrogated its function and therapeutic potential [7,10]. In this study, we developed a first-in-class anti-CSP monoclonal antibody, 1H11, and investigated it as a monotherapy and in combination with standard of care chemotherapies to treat ovarian cancer. Strikingly, 1H11 showed robust anticancer effects as a single agent in in vitro and in vivo OC models. 1H11 caused specific and significant cytotoxicity to OC CSP+ cells and inhibited cell proliferation and migration through the JAK2-STAT3 pathway. In vivo, 1H11 significantly inhibited tumor growth, resulting in 30% tumor necrosis and no observable toxic effects. Moreover, 1H11 synergized with cisplatin and dual treatment caused a sustained reduction in tumor growth compared to cisplatin alone. This study demonstrates the urgent need for future evaluation of CSP and anti-CSP immunotherapy.

Cancer cells regulate the expression of cell surface proteins to drive malignant phenotypes, highlighting the functional importance of the surfaceome as the cell’s signaling gateway and as a target for anticancer interventions [1,3,5,6]. Strikingly, approved therapeutic antibodies only target approximately two dozen cell surface proteins, underscoring the need to widen our cancer cell-surface antigen portfolio [47]. Kelly et al.’s seminal paper revealing the abundant and cancer-specific mislocalization of plectin (CSP) ignited an interest in elucidating the role of noncanonical cell surface proteins in cancer [1]. Since then, plectin has been implicated as a pro-tumorigenic regulator with its ablation resulting in inhibited cell proliferation, migration, and invasion in vitro and reduced tumor volume and metastatic burden in vivo [7,19,20,21,22,23]. Moreover, CSP is emerging as a mitigator of malignant hallmarks. CSP+ cells have increased cell migration, invasion, clonogenicity, and elevated stem cells biomarkers compared to CSP- cells [7,10].

This study expands and mobilizes our understanding of CSP by interrogating the functional consequence of its inhibition with a novel monoclonal antibody. The sequestering of 1H11 to endosomes after internalization suggests that it does not interact with intracellular plectin, instead eliciting CSP-specific targeting repercussions. Moreover, our results showing that 1H11 inhibits cell proliferation and migration in vitro and tumor growth in vivo recapitulate previously reported consequences of CSP modulation [7,10]. Evaluation of 1H11′s molecular mechanism revealed induction of ROS accumulation and inhibition of JAK2-STAT3 signaling, a pathway commonly activated in cancer [48,49,50]. Suppression of the JAK2-STAT3 signaling axis is reported as a promising therapeutic strategy in OC, in part because it regulates the expression of proliferative and migratory effector proteins [43,48,50,51,52]. Consistently, our results showing G0/G1 arrest coincided with an increased expression of cyclin-dependent kinase inhibitors, p27 and p21, while our observed decrease in migration rate correlated with an increase in E-cadherin and a decrease in vimentin expression. Future studies will continue to decipher 1H11′s mechanism of action, differentiating between signaling cascades and phenotypic changes directly modulated by 1H11 from those resulting from signaling crosstalk. Importantly, the induction of inhibitory effects with 1H11 is within the reported therapeutic ranges of targeting conventional functional proteins such as HER2 and EGFR with clinically approved drugs [53,54,55,56]. Overall, not only are we the first to report that therapeutic targeting of CSP is a robust anticancer strategy, thus expanding our anticancer repertoire, but we reveal that targeting a non-conical cell surface protein has molecular consequences. This study underscores the need to further evaluate the biological function and therapeutic relevance of mislocalized cell-surface proteins in cancer. These insights could inform the next generation of novel targets for antibody therapies in cancers.

OC is the deadliest gynecologic cancer, accounting for 150,000 annual deaths worldwide [26]. Under current treatment regimens, OC has an overall cure rate of less than 40% across all stages, illustrating the need for more effective treatment modalities [28]. Predominant OC targeted therapies, consisting of VEGF (bevacizumab) and PARP inhibitors (olaparib), have significant limitations [57,58,59]. For example, anti-VEGF therapy displays toxic side effects such as hypertension and gastrointestinal perforation in OC patients due to VEGF playing a critical role in normal physiological processes [60]. In comparison, CSP is an abundant cancer-specific marker with negligible expression in normal cells, which has resulted in its success as a target for drug delivery systems [1,9]. Although olaparib, the other leading targeted therapy, has progression-free survival benefits, its broad applicability is limited by only ~20% of high-grade serous OC tumors harboring germline or somatic BRCA1/2 mutations [61]. In contrast, Dasa et al. revealed that >90% of serous OC patient samples analyzed were positive for membranous plectin [9]. Thus, compared to current leading therapeutic targets in OC, CSP demonstrates a superior cancer-specific expression pattern. Moreover, since combinatory strategies are often used to augment and prolong efficacy, our results demonstrating 1H11′s strong synergism with olaparib and doxorubicin in vitro and cisplatin in vitro and in vivo have direct clinical implications [29]. Although the complete molecular mechanism underlying these synergistic effects remains to be elucidated, studies have demonstrated that inhibition of JAK2-STAT3 increases sensitivity to cisplatin [62,63]. Furthermore, 1H11′s rapid internalization underscores its potential as an immunoconjugate therapy by being synthesized to drugs, toxins, or radioisotopes to potentiate cytotoxicity. Excitingly, 1H11 as a monotherapy demonstrates a more substantial antitumor effect on OVCAR8 tumor-bearing mice than an HER-3 targeting antibody-drug conjugate [64]. Taken together, our study revealing 1H11′s utility for integration with current mainstay drugs has wide implications on future combinatory cancer strategies in OC.

Beyond OC, our observations provide motivation to investigate the efficacy of CSP-targeted antibody therapy in other CSP+ cancers. Our finding that high plectin mRNA expression is a prognostic indicator for worse overall survival in TP53 mutant serous carcinoma is consistent with previous reports of plectin’s prognostic significance in CSP+ lung adenocarcinoma and head and neck cell carcinoma [8,10,19]. Currently, CSP+ cancers include ovarian, pancreatic, colorectal, bile duct, and lung cancer, which collectively account for more than half of all estimated cancer deaths annually [8,9,11]. Given the wide range of possible applications, 1H11 warrants clinical assessment. Thus, future work will focus on the humanization or chimerization of 1H11 and expand toxicity studies in mice and nonhuman primates. Further evaluation of 1H11 as an oncologic strategy, especially in other CSP+ cancers, could marshal a new era of targeted therapies and patient survival.

This study is the first to reveal anti-CSP therapy as a valuable addition to the current armamentarium of targeted strategies. We demonstrated the potent anticancer activity of CSP-targeting immunotherapy and showed its synergistic interaction with mainstay therapeutics to treat OC. Moreover, not only does our study open the door for further avenues of investigation into CSP, but its bioavailability, cancer-specific expression, and functional significance demonstrate the potential of other noncanonical cell surface proteins to be exploited for anticancer interventions.

## 5. Patents

Plectin-1 binding antibodies and uses thereof: WO2017177199A2 (2017). Kimberly A. Kelly and Julien Dimastromatteo.

## Figures and Tables

**Figure 1 cells-10-02218-f001:**
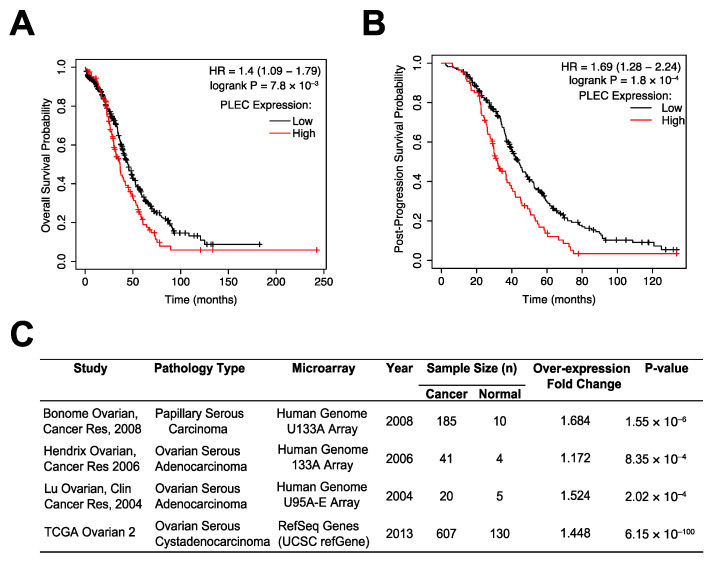
Higher levels of PLEC mRNA expression are associated with poor ovarian cancer patient survival. (**A**,**B**) The KM plotter database was used to evaluate the prognostic value of PLEC mRNA expression in OC’s most common subtype, TP53 mutant serous carcinoma. High PLEC expression is a significant indicator of (**A**) overall survival and (**B**) post-progression survival in this subset of patients. (**C**) A differential expression analysis using Oncomine revealed PLEC to be significantly overexpressed in OC tissues.

**Figure 2 cells-10-02218-f002:**
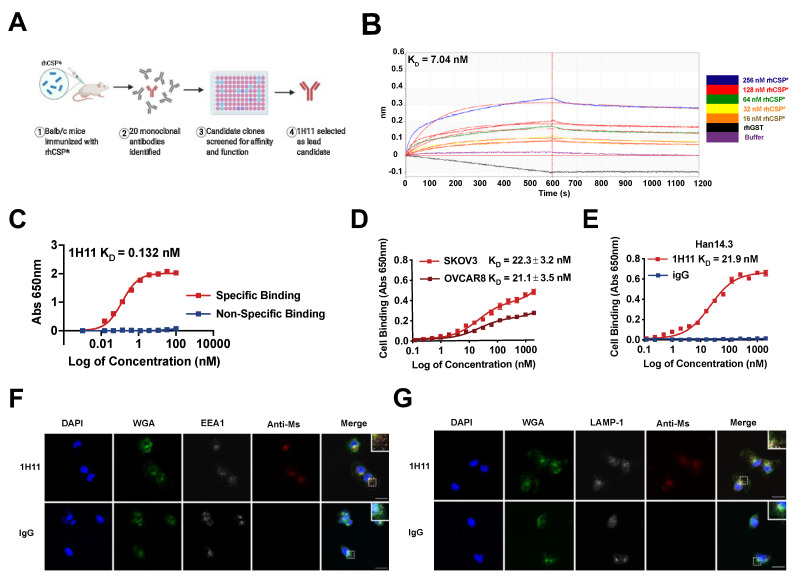
1H11 generation and characterization. (**A**) A workflow schematic of 1H11 development. (**B**) 1H11 shows high binding affinity and specificity to recombinant CSP (rhCSP*) by kinetic binding assay and (**C**) in proteo ELISA. (**D**,**E**) Cell binding assays were performed to evaluate 1H11 binding to human and murine native CSP by incubating (**D**) human CSP+ OC cell lines, SKOV3 or OVCAR8, or (**E**) murine CSP+ cell line, Han14.3, with serial dilutions of 1H11 or IgG. (**F**,**G**) Antibody internalization was evaluated by incubating OVCAR8 cells with 1H11 or IgG for 30 min, then co-staining for markers of endocytosis, (**F**) EEA1 or (**G**) LAMP-1. Immunofluorescence imaging shows rapid internalization of 1H11 compared to IgG. Representative images are shown. Blue, DAPI; green, WGA; white, endosome markers (EEA1 or LAMP1); red, treatment condition (1H11 or IgG). Scale Bar = 25 µm. Errors bars indicate mean ± SEM.

**Figure 3 cells-10-02218-f003:**
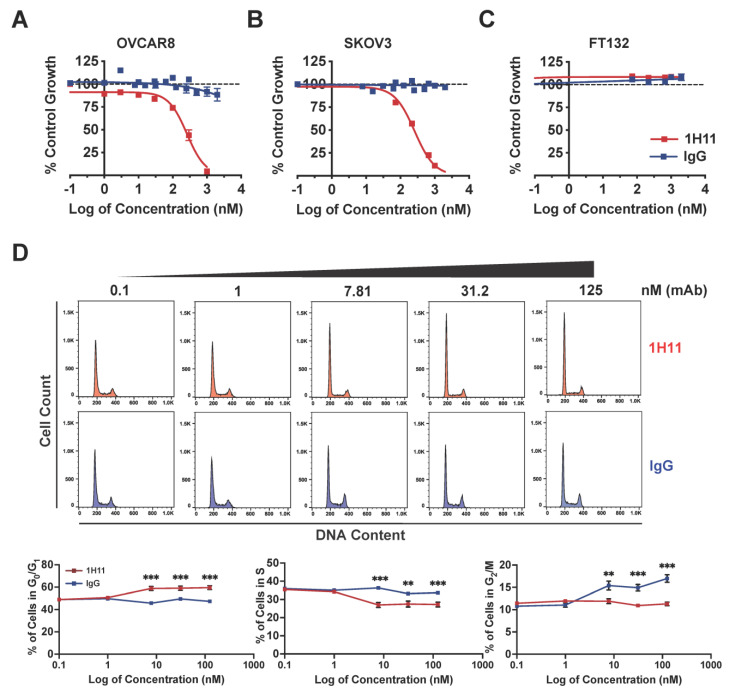
1H11 induces OC cell cytotoxicity and cell cycle arrest. (**A**–**C**) Evaluated by SRB assay, 1H11 selectively inhibited OC cell proliferation with an EC_50_ of 260 nM in both (**A**) OVACR8 and (**B**) SKOV3 with no cytotoxic effects on (**C**) FT132. (**D**) Cell-cycle analysis of OVCAR8 cells treated for 72 h was performed by flow cytometry. Significant G0/G1 cell-cycle arrest was induced in a dose-dependent manner by 1H11. All data plotted as mean ± SEM. *** *p* < 0.001, ** *p* < 0.01 vs. IgG.

**Figure 4 cells-10-02218-f004:**
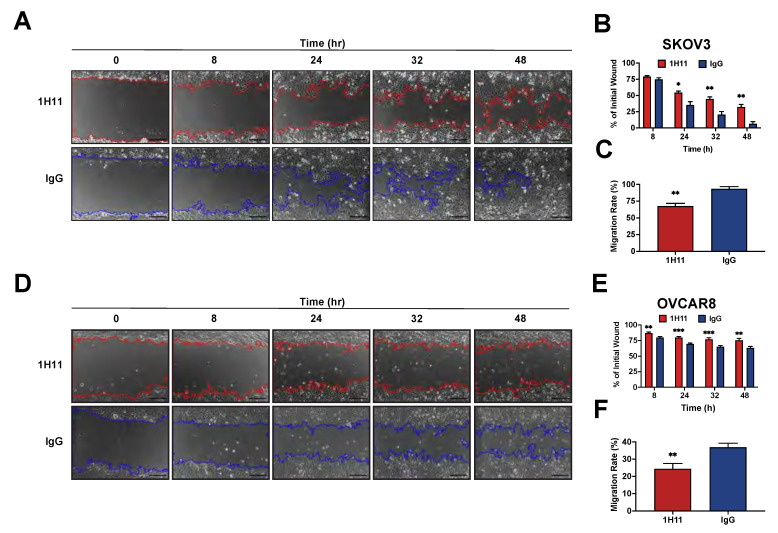
1H11 inhibits OC cell migration. A scratch assay was used to evaluate the migration ability of cells treated with 2 µM 1H11 or IgG. Wounds inflicted on (**A**) SKOV3 and (**D**) OVCAR8 were imaged for up to 48 h. Representative images are shown with a scale bar = 200 µm. (**B**,**E**) Quantification of wound closure demonstrates that 1H11 inhibited cell motility. Similarly, treatment with 1H11 significantly reduced the migration rate of (**C**) SKOV3 and (**F**) OVCAR8. All data plotted as mean ± SEM. *** *p* < 0.001, ** *p* < 0.01, * *p* < 0.05 vs. IgG.

**Figure 5 cells-10-02218-f005:**
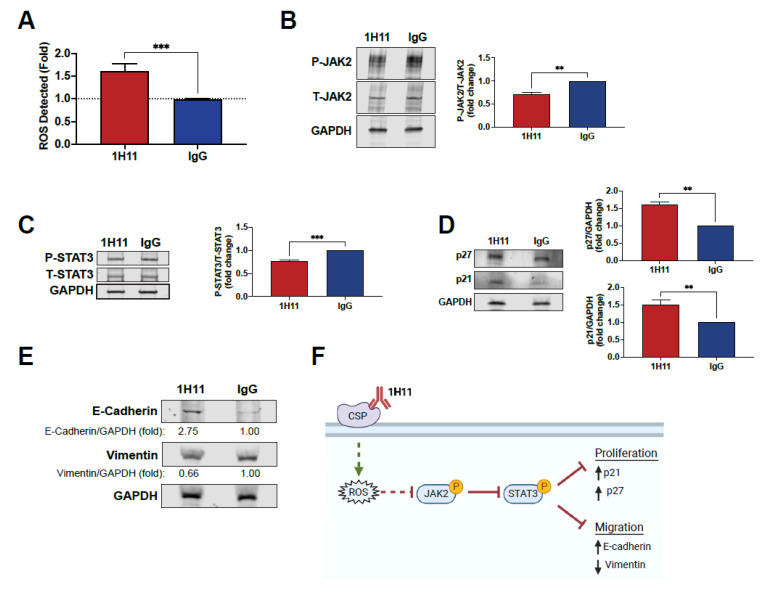
1H11 induces ROS production and modulates the expression of critical proliferative and migratory effector proteins. (**A**) After 72 h of treatment, ROS levels were evaluated by DCFDA staining. 1H11 caused an increase in ROS levels. (**B**,**C**) Serum-starved cells were treated for 10 min, and lysates were accessed by Western blotting. 1H11 treated cells showed decreased phosphorylation of (**B**) JAK2 and (**C**) STAT3. (**D**,**E**) Cells were incubated with 1H11 or IgG for 72 h prior to harvest and evaluation by Western blotting. (**D**) 1H11 treatment resulted in an increased expression of two G1-checkpoint CDK inhibitors, p27 and p21. (**E**) An increase in E-cadherin and a decrease in vimentin protein expression were also observed. For Western blotting, GAPDH was used as a loading control. All experiments were performed by treating OVCAR8 cells with 125 nM 1H11 or IgG. All data plotted as mean ± SEM. *** *p* < 0.001, ** *p* < 0.01 vs. IgG. (**F**) Proposed molecular mechanism of 1H11.

**Figure 6 cells-10-02218-f006:**
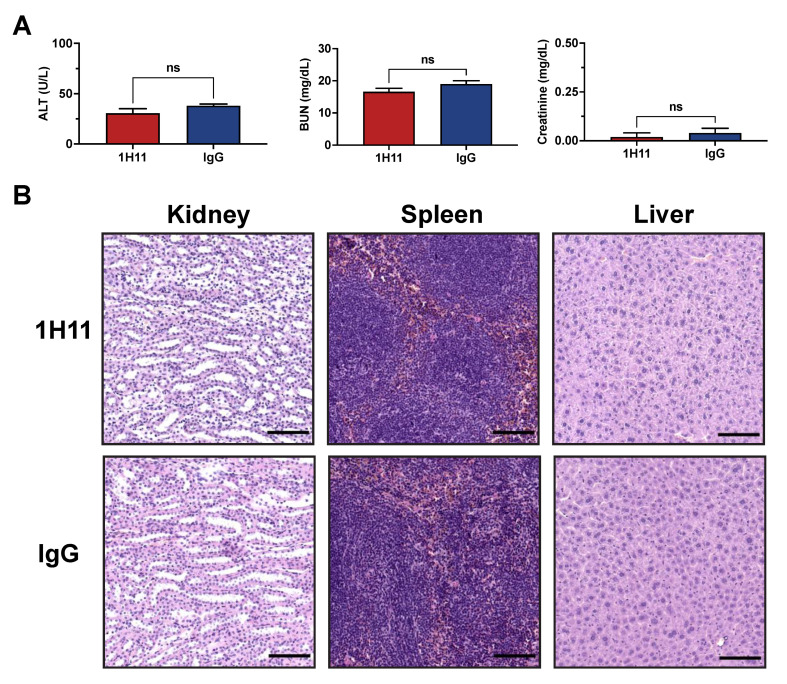
1H11 does not induce major toxic effects. Athymic nude mice with subcutaneously implanted OVCAR8 tumors were treated for one week with 5 mg/kg 1H11 or IgG. (**A**) Chemistry tests for alanine transaminase (ALT), blood urea nitrogen (BUN), and creatinine were conducted from serum samples. No significant changes were observed between groups. All data plotted as mean ± SEM, non-significant (ns) vs. IgG. (**B**) Kidney, spleen, and liver samples stained with H&E showed no major histological difference. Representative images are shown with a scale bar = 100 µm.

**Figure 7 cells-10-02218-f007:**
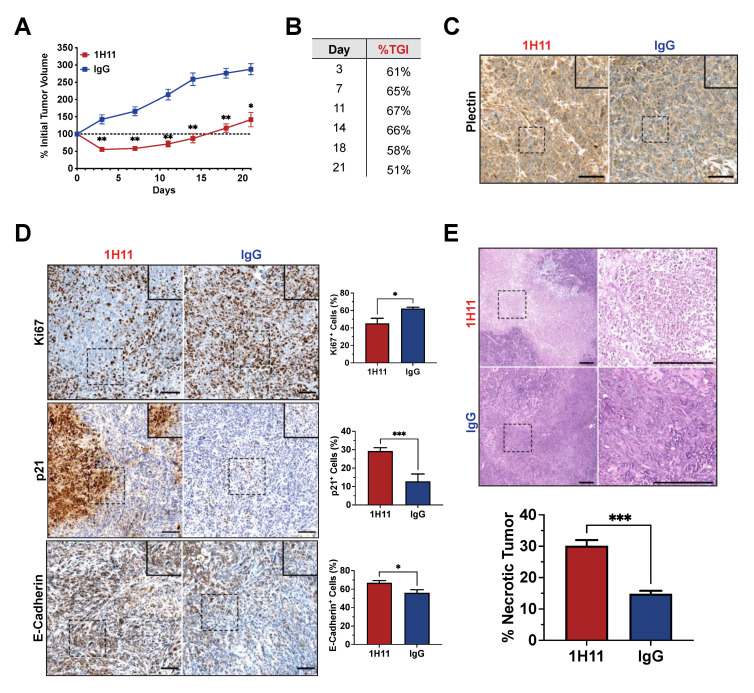
1H11 reduces OC tumor growth in vivo. Athymic nude mice harboring OVCAR8 subcutaneous tumors were treated with 5 mg/kg 1H11 or IgG twice weekly then harvest after three weeks of treatment. (**A**) Tumor growth curves relative to tumor volume at day 0 and (**B**) a table of 1H11′s tumor growth inhibition (%TGI) demonstrates that 1H11 delayed tumor growth. (**C**) Representative images of IHC staining for plectin. Scale bar = 100 µm. (**D**) Representative images of IHC staining and corresponding quantification of Ki67, p21, and E-cadherin. 1H11 prompted an anti-proliferative response. Scale Bar = 100 µm. (**E**) Tumor sections were H&E-stained and analyzed for necrosis. 1H11-treated tumors had a higher percent of necrosis. Representative images are shown. Scale bar = 250 µm. For all images, scale bars are on the lower right, and dashes boxes are zoomed-in regions. All data plotted as mean ± SEM. *** *p* < 0.001, ** *p* < 0.01, * *p* < 0.05 vs. IgG.

**Figure 8 cells-10-02218-f008:**
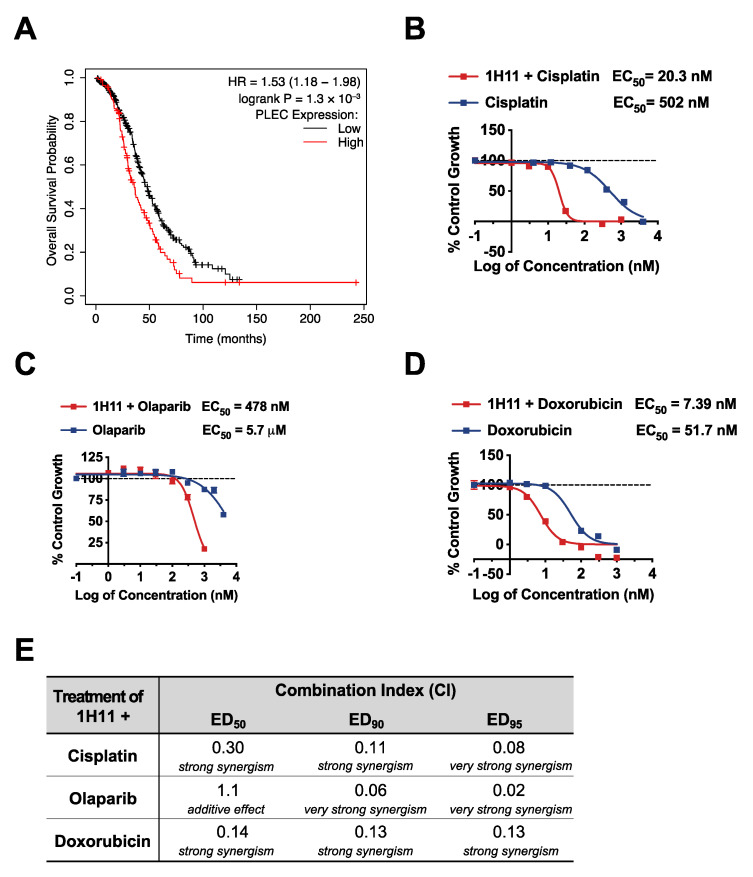
1H11 increases potency of first-line OC therapies in vitro. (**A**) A patient survival analysis using the KM plotter database revealed that in serous mutant TP53 OC patients who received platinum-based chemotherapy, higher plectin mRNA expression is significantly associated with worse overall survival. (**B**–**D**) OVCAR8 cell viability was evaluated by SRB assay. Results demonstrate that dual treatment with 1H11 increases the potency of (**B**) cisplatin, (**C**) olaparib, and (**D**) doxorubicin. All data plotted as mean ± SEM. (**E**) The combination index (CI) values of combinational treatment with 1H11 and different anticancer agents on OVCAR8 cells. The interpretation of CI values is listed below.

**Figure 9 cells-10-02218-f009:**
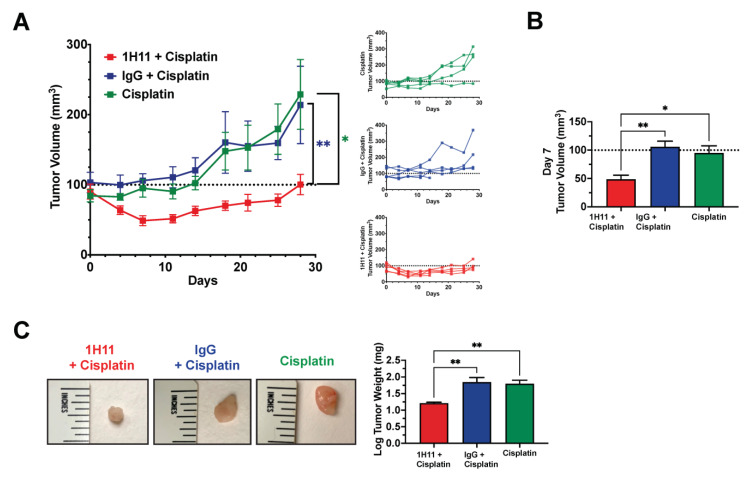
1H11 enhances sensitivity to cisplatin in vivo. Athymic nude mice with OVCAR8 subcutaneous tumors were administered either dual treatment of (1) 1 mg/kg 1H11 and 2 mg/kg cisplatin, (2) 1 mg/kg IgG, and 2 mg/kg cisplatin, or (3) 2 mg/kg cisplatin alone for up to 28 days. (**A**) Average and individual mouse tumor volume curves throughout treatment. (**B**) Tumor volume at day 7 of treatment. (**C**) Tumor weights at the end of the study and representative images of tumors. All data plotted as mean ± SEM. ** *p* < 0.01, * *p* < 0.05 vs. 1H11 + Cisplatin.

## Data Availability

The data presented in this study are available upon request from the corresponding author.

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
