# Peer review of "A Novel Monoclonal Antibody Targeting Cancer-Specific Plectin Has Potent Antitumor Activity in Ovarian Cancer"

_cells, 2021, doi:10.3390/cells10092218_

Round 1
Reviewer 1 Report
The authors are evaluating plectin, which is a structural protein moving on to cell surface in tumors, termed as Cancer-specific plectin (CSP) as a therapeutic target using their plectin specific antibody called 1H11. They performed binding studies, standard in vitro functional studies such as cell viability, cell migration, cell cycle analysis and in vivo tumor burden effects of 1H11 targeting CSP. They also studied 1H11 endocytosis and mechanistic evaluations to investigate the mechanism of action.
Since this study demonstrates the therapeutic utility of targeting noncanonical cell surface protein like plectin, as opposed to conventional “functional” proteins, this opens the door to this class of proteins as novel therapeutic targets. Therefore, this study is significant.
Some concerns that the authors should address before publishing:
- What are the direct evidence on “1H11 did not bind to cytoplasmic plectin”?
- Authors should discuss the need of 2 uM of 1H11 to observe a significant migration inhibition, and EC50 of 260nM, while the binding of 1H11 was around 20 nM. Is this discrepancy due to targeting of unconventional/structural protein as opposed to conventional “functional” proteins? A comparative discussion to typical “functional” protein inhibition such as direct kinase inhibition would help to understand inhibitory behaviors of such noncanonical proteins.
- Why 1H11 treated tumors still show same amount of CSP? Isn’t that get internalized and hence should be reduced?
- The indicated Kelly et al.’s paper needs to be cited in line 418.
Reviewer 2 Report
In this manuscript, the authors newly developed mAb against cancer specific plectin. The mAb showed promising antitumor effect against ovarian cancer through anti-proliferation and anti-migration effect. The research and paper are well-designed, and the findings seem to be interesting. I think the paper is acceptable, but I have just minor comments and questions.
- In 2.3. “Generation and Characterization of 1H11”, the authors describe the methods for preparation of new mAb briefly. But I recommend describing detail information. For example, what is the fragment of human CSP (amino acid sequence from a to b), immunization protocol (dose, schedule).
- How did 1H11 generate ROS after binding to plectin? Also after internalization of plectin, did plectin go back to cell membrane? In in vivo study, the plectin level in tumor tissues was not decreased. It means internalization of plectin following binding of 1H11 do not cause degradation of plectin.
- Did the authors check ADCC effect of 1H11?
